# Photo- and Thermocatalytic CO_2_ Methanation: A Comparison of Ni/Al_2_O_3_ and Ni–Ce Hydrotalcite-Derived Materials under UV and Visible Light

**DOI:** 10.3390/ma16175907

**Published:** 2023-08-29

**Authors:** Rafael Canales, Victoria Laura Barrio

**Affiliations:** School of Engineering of Bilbao, University of the Basque Country (UPV/EHU), 48013 Bilbao, Spain

**Keywords:** CO_2_ methanation, Al–Ce promotion, photocatalytic activity, hydrotalcite, solid solution, catalysts, photocatalyst, hydrotalcite-derived materials

## Abstract

Catalysts derived from Ni/Al/Mg/Ce hydrotalcite were prepared via a co-precipitation method, varying the Ce/Al atomic ratio. All of the catalytic systems thus prepared were tested for CO_2_ methanation under dark and photocatalytic conditions (visible and ultraviolet) under continuous flow with the light intensity set to 2.4 W cm^−2^. The substitution of Al by Ce formed a solid solution, generating oxygen vacancies and Ce^3+^/Ce^4+^ ions that helped shift the dissociation of CO_2_ towards the production of CH_4_, thus enhancing the activity of methanation, especially at lower temperatures (<523 K) and with visible light at temperatures where other catalysts were inactive. Additionally, for comparison purposes, Ni/Al_2_O_3_-based catalysts prepared via wetness impregnation were synthesized with different Ni loadings. Analytical techniques were used for the characterization of the systems. The best results in terms of activity were as follows: Hydrotalcite with Ce promoter > Hydrotalcite without Ce promoter > 25Ni/Al_2_O_3_ > 13Ni/Al_2_O_3_. Hydrotalcite, with a Ce/Al atomic ratio of 0.22 and a Ni content of 23 wt%, produced 7.74 mmol CH_4_ min^−1^·g_cat_ at 473 K under visible light. Moreover, this catalyst exhibited stable photocatalytic activity during a 24 h reaction time with a CO_2_ conversion rate of 65% and CH_4_ selectivity of >98% at 523 K. This photocatalytic Sabatier enhancement achieved activity at lower temperatures than those reported in previous publications.

## 1. Introduction

Most global activities use energy sources to operate; indeed, the products associated with manufacturing processes are processed in ways that increase global CO_2_ emissions. Thus, there is an increasing demand for technologies that are able to mitigate global warming. This worrying increase in emissions due to increases in population and industrial demand has generated investigations into different ways of reducing these emissions. For example, advanced techniques such as methanation [1,2,3] or CO_2_ capture [4,5] are continuously evolving. The primary advantage of CO_2_ capture is its potential to reduce greenhouse gas emissions, particularly CO_2_, which is the primary driver of climate change and could be integrated with existing industrial infrastructure. However, operational costs are still too high, and further developments are needed. In the case of membrane technology, it has shown promise in addressing challenging separations, including the removal of carbon dioxide (CO_2_), sulfur dioxide (SO_2_), and nitrogen oxides (NOx) from flue gases and waste gases. These membranes selectively allow CO_2_ to permeate while blocking other gases like nitrogen or oxygen. The captured CO_2_ can then be recovered for storage or utilization purposes.

Firstly, the advantages of different membrane-based CO_2_ systems with high permeability are based on different materials. Polymer membranes, which are relatively inexpensive to produce and can be easily scaled up, have the possibility of operating at low temperatures, which can lead to energy savings, and their microporous structure to high permeability, enhancing solubility and selectivity for CO_2_ (14.3 of CO_2_/N_2_ selectivity). Mixed matrix membranes (MMMs) improved some properties of the polymer membranes by adding filler materials into the polymer membrane, improving affinity, diffusivity, mechanical stability, and CO_2_/N_2_ selectivity by 68%. Finally, the facilitated transport membranes enhance 90% CO_2_/N_2_ selectivity and more than 50% CO_2_ permeability due to their high CO_2_ permeability and potential for energy efficiency operating at lower temperatures and pressure differentials compared to traditional methods.

The main drawbacks of polymeric membranes are limited chemical resistance, temperature and pressure constraints, plasticization, and membrane aging. The disadvantages of mixed matrix membranes are particle sedimentation, interfacial voids, and filler–polymer incompatibility. Finally, the facilitated transport membranes have low stability, carrier leakage, and saturation of the carrier [4].

Lakshminarayana Bhatta et al. studied promising materials using hydrotalcite materials combined with metal oxides for CO_2_ capture. Hydrotalcites are superior to other materials due to their high selectivity for CO_2_ chemisorption and good regenerability. The number of basic strength sites for improving CO_2_ sorption can be modified by controlling the number of defects in the material, like adding zeolites. The sorption capacity (wt% of CO_2_) combining hydrotalcites coated with zeolites was 30.8% using 100% of pure CO_2_ if they are compared with other promoters such as K–Na (5.3%) or non-modified hydrotalcites (7.0%). The major disadvantages are the adverse effect of water vapor on sorption stability and the high cost of the material due to zeolites’ addition [5].

Additionally, M.R. Rahimpour et al. studied the different effects of CO removal connected with hydrogen permselective membrane reactors on methanol production with high CO conversion and H_2_ permeation rates [6]. Further research is needed, such as the combination of CO_2_ at low pressure and its subsequent hydrogenation process for the production of methanol [7].

After the CO_2_ separation, CO_2_ conversion proceeds. This work focused on improving thermochemical and photochemical catalysts using CO_2_ and transforming it into highly stable CH_4_. Moreover, the current natural gas pipelines should be capable of transporting methane, unlike the storage and transport of H_2_, which needs further research as only a small percentage is allowed in the natural gas network. The high pressure and energy needed to store H_2_ are other fields that researchers have focused on, as are the materials used [8,9]. Power-to-gas technology uses excess renewable electricity to produce H_2_, normally via the electrolysis of water [10,11]. Therefore, the H_2_ generated can be considered in two different ways: as a final energy product for storage/distribution or, as in this work, as a chemical reactant.

This study focused on the Sabatier reaction (Equation (1)), which is a highly exothermic reaction at mild temperatures (523–723 K). Other reactions (Equations (2)–(4)) that can happen are the following [12]:
CO_2_ + 4H_2_ ⇄ CH_4_ + 2H_2_O ΔH_298 K_= −165.0 kJ/mol(1)
CO_2_ + H_2_ ⇄ CO + H_2_O ΔH_298 K_ = −41.2 kJ/mol(2)
CO_2_ + 2H_2_ ⇄ C + 2H_2_O ΔH_298 K_ = −90.1 kJ/mol(3)
CO + 3H_2_ ⇄ CH_4_ + H_2_O ΔH_298 K_ = −206.1 kJ/mol(4)

Table 1 provides a short summary of the materials investigated and the reaction conditions applied. As can be observed, the most common supports are alumina, titania, and ceria, and the main metals are gold, ruthenium, and nickel, among others, normally at temperatures between 523 and 573 K. The use of noble metals must be avoided because of their price and availability. Regarding precedents in terms of the reaction mechanism, the role of the plasmon resonance band, the average particle size, and the replacement of metal oxides acting as support, the valence–conduction band energy and the CO_2_ adsorption capacity are the main characteristics that enhance the activity of the dark catalytic and photocatalytic methanation of CO_2_.

The key goal of photocatalytic reactions is to achieve continual improvements in the catalysts’ sensitivity to sunlight. As seen in Appendix A, most solar radiation energy is in the visible region, so one desirable innovation is the use of photosensitive catalysts in this region [20,22]. In our study, we developed catalytic systems that present a different band gap, which was confirmed by their light absorbance and reflectance (Tauc plots).

Currently, hydrotalcite-derived materials (HTCs) have several uses in the field of fuels, such as CO_2_ capture and storage, the conversion of natural gas, the mitigation of pollution, etc. As natural minerals (Appendix A), hydrotalcites belong to a class of anionic and basic clays with a general structure of [$M_{1-x}^{2+}M_{x}^{3+}({OH)}_{2}$] (A^n−^)_x/n_ z$H_{2}O$, in which M represents the divalent or trivalent metal cations, A^n−^(CO_3_^−^, Cl^−^, NO_3_^−^) represents the anions of the interlayer, and x (M^3+^/M^3+^ + M^2+^) is the molar ratio, which is normally fixed between 0.25 and 0.33. The partial substitution of divalent or trivalent cations with a different nature in HTCs can modify the structure, generating interesting properties. These properties include the modification of the specific surface area, the addition of structural defects, the enhanced dispersion of active metals, and improvements in acidic–basic structural sites. These modifications might enhance the hydrogenation of CO_2_, as listed in Table 1. The roles of Ni and CeO_2_ in the Sabatier reaction have been frequently described, but the addition of ceria as a promoter in the synthesis of hydrotalcites for photocatalytic CO_2_ methanation has seldom been discussed [23]. Therefore, in this study, the atomic ratio of Ce/Al was changed from 0 to 0.22, considering Ce to be a trivalent metal and fixing the Ni content at 25 wt.%. The catalysts were analyzed under three different reaction conditions: in the dark, under ultraviolet light, and under visible light. In addition to the molar substitution, we explored the interaction with other metals, such as Mg, which positively influenced the methanation of CO_2_ due to its basic properties. Quan Luu Manh Ha et al. observed that the formation of oxygen vacancies and cationic Ce^3+^ defects were two key structural factors that might enhance the activation of CO_2_ [24,25].

Moreover, Ni/Al_2_O_3_ samples with different Ni contents were synthesized to compare their activity with samples prepared with HTCs. Firstly, a catalyst of 13Ni/Al_2_O_3_ was used, in line with the previous results achieved by David Méndez et al. [1]. The method, owing to the Ni, showed good performance with the loading described. However, the aim of this study was to analyze the catalytic performance by varying the Ni content on γ-Al_2_O_3_ from 13 to 25% for comparison purposes, analyze the charge with the best thermal and photothermal activity results, and proceed to a comparison with hydrotalcites with the same nickel content. In this study, the photocatalysis of the traditional Sabatier reaction was investigated using two different light sources: ultraviolet (UV) and visible light. The results were compared with those obtained from the conventional thermocatalytic process. Hydrotalcite-like materials containing different loadings of Ce and Ni prepared via a previously reported co-precipitation method were used as the active metal catalysts.

## 2. Experimental Section

### 2.1. Preparation of the Catalysts

Hydrotalcite-derived materials containing Mg, Al, Ni (HTC 1), and different loadings of Ce (HTC 2-3-4) were synthesized via the co-precipitation method [26]. With Ce as a trivalent metal, an amount of Al was replaced by the addition of Ce in order to maintain the same concentration of trivalent metals. The molarity was fixed by maintaining the molar ratio of M^2+^/M^3+^ equal to 3 (or M^3+^/(M^2+^ + M^3+^) = 0.25). This ratio is also seen in natural hydrotalcite: Mg_6_Al_2_CO_3_(OH)_16_·4H_2_O. The metal precursors used are detailed in the Appendix A. A 1 M solution of metal salt diluted in Milli-Q water was added to a beaker. The nominal weight content of the components is listed in Table 2.

In order to maintain a pH of 10 ± 0.2, solutions of 2 M NaOH and 0.125 M Na_2_CO_3_ were prepared, added dropwise, and continuously agitated at ambient temperature. The resulting solution was pressure-washed and filtered until a pH of 7 was reached. The paste was kept overnight at 373 K and was calcined afterwards at 723 K (5 K/min for 4 h). Moreover, the other Ni-based catalysts supported on two different γ-Al_2_O_3_ were prepared by the wet impregnation method [1] with the same nickel nitrate salt. Two metal loadings of Ni (13 wt% and 25 wt%) were added to the solution dissolved in distilled water. Excess distilled water was added to achieve an optimal pH with the addition of ammonium (Panreac) [1]. After 12 h of stirring at a constant pH, the sample was evaporated with an evaporator (Heidolph Laborata 4000, Schwabach, Germany) connected to a vacuum pump.

### 2.2. Activity Tests

The samples were initially pressed at 8 tons and then sieved to pass through two different sieves in the range of 0.42–0.5 mm. The reaction was performed in a photoreactor (Figure 1A) with an open removable quartz window of 8 mm at 53.5 h^−1^. The bench-scale plant (PID Eng&Tech, Madrid, Spain) had an LED light source attached to the window. The composition of the effluent gases was analyzed by an online CompactGC 4.0 gas chromatograph with a TCD and an FID. The intensity of the LED lights was analyzed by a GL Optic Spectis 1.0 spectrometer coupled to an Opti Sphere 48 (GL Optic, Puszczykowo, Poland).

Depending on the sample, the reduction protocol was different. In one case, for the monometallic Ni/Al_2_O_3_ samples, in situ reduction was performed with a H_2_ mixture (99.999%) with a N_2_/H_2_ ratio of 1. In the case of the HTCs, ex situ reduction was performed due to the high reduction temperature needed with a flow of 30% H_2_/N_2_, as shown in Figure 2. Once the sample had been reduced according to the maximum temperature peak in the TPR-H_2_ results, the system was heated with N_2_ from room temperature to the desired temperature (ramp rate: 5 K/min) at 9 bars. For each type of activity test, the reaction was executed in a dark environment with a white LED (365 nm, 3.4 eV) and a blue LED (470 nm, 2.7 eV) under well-controlled lighting conditions, as presented in Figure 1B. For the activity tests, the temperature was changed across the range of 473–725 K, the system was maintained at each temperature for 2 h, and the outlet gases were continuously analyzed. Once the target temperature had been reached, N_2_ was used to heat (5 K/min) and purge the gases inside the plant until the next activity test. Activities at temperatures below 473 K were not analyzed due to the possibility of water condensation.

The activities of the gas phase, such as the CH_4_ yield and the conversion of H_2_ and CO_2_, were calculated using the following Equations (5)–(7):(5)$${CO}_{2}\;Conversion\;(\%)=\frac{\left[{CO}_{2}\right]in-\left[{CO}_{2}\right]out}{\left[{CO}_{2}\right]in}\times 100$$
(6)$$H_{2}\;Conversion\;\left(\%\right)=\frac{\left[H_{2}\right]in-\left[H_{2}\right]out}{\left[H_{2}\right]in}\times 100$$
(7)$${CH}_{4}\;yield\;\left(\%\right)=\frac{\left[{CH}_{4}\right]out}{\left[{CO}_{2}\right]in}\times 100$$

## 3. Results

### 3.1. Characterization of the Catalysts

The characterization procedure to analyze the physico–chemical properties of the samples is detailed in Appendix A.

#### 3.1.1. Inductively Coupled Plasma Optical Emission Spectroscopy (ICP-OES)

The elemental composition of the prepared catalysts was determined by spectrometric ICP-OES after acid digestion. The metal elements determined by this analysis are presented in Table 3. The calculated molar ratios were almost equal to the nominal ones, namely, M^3+^/(M^2+^ + M^3+^) = 0.25. The increase in the Ni/(Mg + Al) ratio was due to the replacement of Al by Ce; this last metal was considered to be Ce^3+^.

#### 3.1.2. Brunauer–Emmett–Teller (BET)

For all hydrotalcites, the isotherms obtained were Type IV, which are typical for mesoporous materials. HTC 1 had the lowest surface area but the highest pore diameter. By increasing the Ce content and decreasing the Al content, we observed a decrease in the surface area. This behavior can be seen in Table 3, in which HTC 2 had an increased surface area compared with HTC 1, but the addition of more Ce slightly decreased the surface area. Furthermore, the surface area of pure γ-Al_2_O_3_ (268 m^2^/g) decreased slightly compared with 13Ni/Al_2_O_3_ (178 m^2^/g) with the addition of Ni. Two different types of γ-Al_2_O_3_ supports were used, and the textural properties of each support are given in Table 3. In the case of 13Ni, a support of 202 m^2^ g^−1^ with a pore volume of 0.81 cm^3^/g and a pore diameter of 7.7 nm was synthesized. Characteristics of the 25Ni sample are shown in Table 3.

#### 3.1.3. The Programmed Reduction Temperature of H_2_

For the calcined HTCs, the H_2_-temperature-programmed reduction (H_2_-TPR) profiles are shown in Figure 2A. For HTC 1, two asymmetric peaks were observed from 600 K to 680 K, with a third one at 1035 K. The first consumption peak was attributed to the reduction in NiO species with a weak interaction with a solid solution or the Ni-phase reduced by free NiO species [27,28]. In the case of the hydrotalcite with the Ce promoter, the peaks that appeared at the lower temperature between 540–550 K and 650 K were related to the NiO species that weakly interacted with the surface CeO_2_ species [29]. The green peaks (<990 K) were associated with the interaction of the Ni^2+^ species with the solid solution or with the combination of Ni^2+^ species with a non-stoichiometric spinel, which strongly interacts with the support. The blue and red peaks represent the reduction in the bulk NiAl_2_O_4_, and this reduction was more intense for higher peaks in the alumina contents. The more intense blue peak for HTC 4 and its shift to a lower reduction temperature compared with HTC 2 and HTC 3 were due to its higher Ce/Al ratio. This ratio was related to the higher CeO_2_ bulk content and the lower NiAl_2_O_4_ reduction [30].

It was also observed that the reduction temperature tended to decrease with the introduction of Ce. Nevertheless, changing the loading of Ce did not seem to strongly affect the reducibility of the samples.

For the Ni/Al_2_O_3_ samples (Figure 2B), the Ni^2+^ was completely reduced to Ni^0^. Thus, the H_2_-TPR profile for the 13Ni/Al_2_O_3_ catalyst suggests a strong interaction of the NiO species with the support corresponding to the red and blue peak [31]. With a higher Ni content (25Ni/Al_2_O_3_), higher H_2_ adsorption was measured, with a shoulder centered at 675 K (green peak), which was related to larger particles that could easily be reduced.

#### 3.1.4. CO_2_-Temperature-Programmed Desorption (CO_2_–TPD)

The CO_2_-TPD basicity profiles of the HTC materials of the calcined samples with 10% CO_2_/He are represented in Figure 3. The basic sites presented in Table 4 were calculated by integrating the areas of weak, medium, and strong basic sites per gram of catalyst. Weak basic centers with Brønsted hydroxyl groups were measured between 353 and 473 K; medium basic centers were associated with Lewis acids and bases between 473 and 623 K; and strong basic sites were related to less coordinated O^2−^ anion centers between 623 and 773 K [32]. The first desorption peak corresponded to bicarbonate, while the medium and strong basic centers were related to bidentate carbonate and unidentate carbonate, respectively [33]. All HTCs showed high CO_2_ adsorption at moderate sites (44–49%). However, the highest CO_2_ adsorption was found for HTC 1 and 2 (with the highest MgO content) and HTC 4 (with the highest CeO_2_ content). The addition of Ce did not change the adsorption profiles; thus, the replacement of Al by Ce also led to an increase in the basicity due to Ce and its introduction of oxygen vacancies in the structure that led to the dissociative adsorption of CO_2_ [34]. Higher desorption temperatures were not studied due to the decomposition of the hydrotalcites and the instability of the catalyst while the material’s properties changed. In earlier analyses of monometallic Ni/Al_2_O_3_ catalysts, a strong interaction was observed due to the contribution of Ni to strong sites [31].

#### 3.1.5. X-ray Diffraction (XRD)

The XRD patterns used for analyzing the crystalline species of non-calcined HTCs are provided in Figure 4A. The basal (0 0 3) and non-basal (1 1 0) reflections at 2θ 11.5° and 60.5°, respectively, were characteristic of the structure of hydrotalcite (ICDD 14–191). The equation that relates the net broadening of the diffraction peak (β) to the size of the crystallite (τ) is the Scherrer equation (Equation (8)), in which K is 0.89 (theoretical spheres) and λ is the X radiation wavelength. The cell parameter “c” indicates the expansion caused by the hydroxyls and carbonates compared with natural hydrotalcites. The data for the crystal size were collected in Table 5.
(8)$$T=\frac{K\times \lambda }{\beta \times cos\;\theta }$$

Moreover, when the sample was calcined and reduced (Figure 4B), the peaks were wider than those of the non-calcined samples. This means that the longer the sample is calcined, the more amorphous it is. For the alumina, the amorphous phase was not observed [35]. The crystal sizes of the calcined and reduced samples due to the peaks of NiO (ICDD 47–1049) and MgO (ICDD 45–946) (with the same space group and similar cell parameters) were similar. Occasionally, they are referred to as MgAl_2_O_4_ (ICDD 82–2424) and/or NiAl_2_O_4_ (ICDD 71–0963) [36], but for these samples, these peaks were not observed. However, the XRD patterns confirmed that calcination at 723 K promoted the migration of Ni^2+^ ions from the surface to MgO, thus creating a NiO–MgO solid solution [37]. Another aspect is that the Ni metal was partially reduced (ICDD 4–850). For the promoted catalysts, the diffraction lines of the CeO_2_ phase (ICDD 34–394) could be observed.

Finally, the XRD diffraction lines for the 25Ni/Al_2_O_3_ catalyst were also analyzed (Figure 4C). For the 25Ni/Al_2_O_3_ catalyst, three different XRD patterns were obtained: first for the reduced catalyst at 723 K and then for the catalyst after being used in the dark reaction and the UV photocatalytic reaction. Under all conditions of analysis, the diffractograms indicated the presence of a Ni^0^ crystal structure (JCPDS 00-004-0850). Furthermore, the other peaks were ascribed to γ-Al_2_O_3_ (JCPDS 01-079-1557) and NiO (JCPDS 00-047-1049).

#### 3.1.6. H_2_ Chemisorption

The metallic surface area, dispersion, and diameter of active particles for the freshly calcined catalysts were measured and are compiled in Table 6. The stoichiometric factor for Ni was H_2_/M = 0.5. The maximum value of the metallic surface area was 4.13 m^2^·g_cat_^−1^, which was achieved for HTC 2. For higher Ce/Ni ratios, the dispersion of metal decreased and depended on the Ni and CeO_2_ species on the surface due to the reducibility of both species. In the case of Ni supported on Al_2_O_3_, a larger area achieved a greater dispersion of metal.

#### 3.1.7. X-ray Photoelectrons Spectroscopy (XPS)

XPS characterization was used to identify the oxidation states of the main metals. The surface elemental composition was analyzed to a depth of 10 nm below the surface using XPS.

According to Table 7, the XPS results of the Ni 2p, Ce3d, and Al 2p energy stated for the reduced catalysts were analyzed to compare the variation in the metals. It can be clearly observed that the Ni/Al ratio calculated by XPS was lower than the bulk measured by ICP-OES characterization. This difference might be due to the heterogeneous distribution of Ni and Ce dispersed in the support or to the solid solution of NiO–MgO, where the Ni^2+^ ions accumulated in bulk and not on the surface [38].

For all of the HTCs and monometallic Ni/Al_2_O_3_ catalysts, the Ni 2p spectra are presented in Figure 5A. Three main peaks were observed for the following species: Ni^0^, Ni^2+^ from NiO, and NiAl_2_O_4_ and its satellite [39]. Regarding the binding energies obtained from the XPS spectra of the fresh and reduced samples, the HTCs showed three different peaks at around 852.8, 856.8, and 862.9 eV. These values were higher than the ones achieved for 25Ni/Al_2_O_3_.

The O 1s core level presented in Figure 5C presented a single peak at ≃530.9 eV that is representative of the hydroxyl environment. This peak presents two well-defined deconvolution components at around 529.7 and 531.4 eV. The first peak is due to O^2−^ ions present in the solid solution, while the second one is due to the oxygen ions in the solution doped with Al [21]. The Al 2p/Ni 3p core level is presented in Appendix A. The peak at 74.1 eV corresponded to Al_2_O_3_ [40].

Moreover, the spectrum of Ce 3d was analyzed (Figure 5D). HTCs with a Ce promoter presented the Ce^3+^ and Ce^4+^ core levels listed in Appendix A. The four peaks are labeled as x_2_ and x_4_ and as x_1_ and x_3_, corresponding to the transitions of Ce from Ce^3+^ to Ce^4+^, respectively. However, upon deconvolution of the peak fitting, it was determined that HTC 4 exhibited reduced/oxidized fractions of 39.8% Ce_2_O_3_ and 60.2% CeO_2_, while HTC 3 had 33.6% Ce_2_O_3_ and 66.4% CeO_2_. These ratios were similar to the ratios reported for reducibility [39]. Ce can act as a doping element because of the alteration of Ce^4+^/Ce^3+^, which accelerates the separation of charge and thus improves the catalytic reaction on the surface [41]. These electrons are trapped in the oxygen vacancy of the conduction band, leading to photocatalytic enhancement in the visible region [42].

#### 3.1.8. UV–vis Diffuse Reflectance Spectroscopy (UV–vis–NIR DRS) and Band Gap

The UV–vis spectra of the calcined catalysts indicated the adsorption of light at different wavelengths, as shown in Figure 6. The most intense region was in the UVA region (315–400 nm), corresponding to the energy range of 3.10 to 4.13 eV. Monometallic Ni/Al_2_O_3_ exhibited strong absorption from 240 to 370 nm, which was related to the O^2−^ → Ni^2+^ charge transfer transition. For the hydrotalcite materials, this range was attenuated, and the range was extended to 420 nm. The peak intensity for the HTCs was also related to the presence of cerium oxide in both oxidation states (Ce^3+^ and Ce^4+^) and the Ce–Al interaction. Moreover, according to the literature, the highest peak of absorption at 310 nm was attributed to the O^2−^ to Ce^4+^ charge transfer. The shoulder between 590 and 700 nm and the peak at 650 nm suggested the presence of Ni^2+^ ions, which were related to nickel aluminate. This was more intense for a lower Ce/Al ratio and a higher Ni content (13 vs. 25Ni/Al_2_O_3_) [43].

For photocatalysis, it is essential to know the materials’ behavior when they are exposed to light, so calculating the band gap via Tauc plots is very important [44,45]. In this case, two different pathways (direct and indirect) were studied in order to obtain the band gaps. The absorption was linearized by the function (αE)^2^, where α is the absorption coefficient (Kubelka–Munk) obtained from the diffuse reflectance spectroscopy transformation and E is the energy (eV) related to the adsorption wavelength.

In a comparison of the different band gaps, all of the samples strongly absorbed light from the UV (365 nm) range. However, beyond 423 nm, the catalysts tended to be transparent, as the photoexcitation of the semiconductors under UV illumination resulted in an accumulation of electrons in the metal from the shift in the surface plasmon resonance band of the Ni nanoparticles [46]. Moreover, monometallic catalysts of alumina and hydrotalcite with Ni, Al, and Ce metal fillers from previous literature obtained band gaps similar to those in this study [47,48].

Finally, the addition of Ce improved the absorption of the visible region, which could enhance the photocatalytic activity. This contrasted with the theory, and the red shift in the visible region that resulted from the absorption of Ni NPs was due to the surface plasmon resonance effect as a consequence of the improvement produced by using HTCs with higher Ce content.

### 3.2. Performance of the Catalysts

The activity levels, expressed as CO_2_ conversion and CH_4_ yield for HTC 4 at 523 K using visible light, are presented in Figure 7A–D. For the dark reactions, all catalysts exhibited activity at 575 K. For the HTCs, different results were observed with respect to the 13Ni and 25Ni/Al_2_O_3_ catalysts, as they presented higher activities at lower temperatures, mainly due to the interaction of different metal oxides and also because of the substitution of Ce. Indeed, the major contribution of the medium-strength basic sites of the HTCs was provided by Ce and Mg, as well as the oxygen vacancies and Ce^3+^ ions, which were highly effective for the reaction via a redox reaction and also as an active interphase near the metals’ active sites.

The catalytic performance of CO_2_ methanation with Ni–Ce catalysts has been previously investigated. Yanyan Feng et al. compared Ni–Ce catalysts supported on alumina and carbon nanotubes at 623 K [49]. While the addition of ceria improves the catalytic activity and stability compared to catalysts without a promoter, the importance of the support must be emphasized with a CH_4_ yield of 83%. In the previous literature, the importance of moderately basic centers in addition to hydrogen storage capacity has also been discussed. In our study, the catalyst with the highest ceria content was already active at 573 K with visible light. At lower temperatures, a two-phase reaction could occur due to the possibility of water condensation.

On the other hand, Min-Jae Kim et al. conducted a study varying the ceria content. They found that a catalyst with 15% Ni and 15% Ce had the highest activity but was far from the equilibrium reaction, with a maximum CO_2_ conversion rate of 70% at 623 K and high stability at 80 h but almost no activity at 473 K [50].

Regarding the Al_2_O_3_-supported catalysts, the Ni content (13 vs. 25 wt%) had a significant influence at temperatures between 575 and 625 K, enhancing the activity under the higher Ni content, i.e., 25Ni/Al_2_O_3_, in contrast to that under 13Ni/Al_2_O_3_, which showed less activity up to 725 K.

Regarding the reaction enhanced by UV light (Figure 7B), for all catalysts, the activity was higher compared with the performance in the dark and was higher at much lower temperatures. HTC 3 and HTC 4 exhibited high activity, with CO_2_ conversion rates of 66% and 75%, respectively, at 523 K, while the others did not present activity until 573 K.

Regarding the activity enhanced with visible light (Figure 7C), the activity started at lower temperatures compared with UV light. The main difference was observed at 473 K, where CO_2_ conversion rates of 39% and 48% were achieved for HTC 3 and HTC 4, respectively, while these were not active with UV light (summarized in Table 8). Thus, as measured with the UV–vis spectra, the higher absorbance of visible light for the Ce-doped catalyst implies greater photon adsorption (Ce charge transfer), which enhanced the CO_2_ methanation activity [51]. Moreover, the substitution of Ce added some new energy levels below the conduction band, making this hydrotalcite-like precursor more active in the visible range [52]. These electrons were trapped in the oxygen vacancy in the conduction band, allowing photocatalytic enhancement in the visible region. These properties of Ce as a promoter show that this is a successful catalyst for photocatalytic CO_2_ methanation due to the capacity of Ce to switch between Ce^4+^ and Ce^3+^ [42]. In Appendix A, a comparison of methane yield and production versus temperature has been made in visible light.

Finally, the stability of HTC 4 for 24 h under visible light at 523 K was investigated. As shown in Figure 7D, excellent CO_2_ conversion (black square) and CH_4_ yield (blue square) were measured without deactivation.

According to the photocatalytic mechanism for the enhanced methanation of CO_2_, the photons’ energies and the phenomenon known as localized surface plasmon resonance (LPSR) were responsible for this improvement. Normally, this behavior is attributed to noble metals such as platinum and gold, but recent works have included Ni as a plasmonic metal [53,54]. Semiconductors can be activated under light irradiation when E_bandgap_ ≤ E_photon_. However, the system could be unstable due to the recombination of photo-excited electron–hole pairs, leading to unfavorable photocatalytic effectiveness. This light might activate the metal and/or the support, leading to an eight-electron reduction reaction to produce CH_4_: 8H^+^ + 8e^−^ + CO_2_ → CH_4_ + 2H_2_O; and two electrons for CO production: 2H^+^ + 2e^−^ + CO_2_ → CO + H_2_O [55].

In the tests of CO_2_ hydrogenation activity, the carbon balance was between 96.0 and 99.0%. Thus, under the indicated operating conditions, the formation of other species, such as HCOOH (2e^−^), HCHO (4e^−^), and CH_3_OH (6e^−^), was not observed [56].

#### Scanning Transmission Electron Microscopy Using Energy Dispersive X-ray Spectroscopy (STEM–EDS) and XPS of the Used Catalysts

The STEM micrographs showed a highly porous metal oxide structure of an HTC-like precursor prepared by co-precipitation. The Ni NPs were well dispersed with a homogeneous particle size in all samples, in which the particle sizes ranged between 8 and 10 nm. For the 13 and 25Ni/Al_2_O_3_ catalysts, larger particles were observed. Moreover, the 25Ni/Al_2_O_3_ catalyst had larger particle sizes than the 13Ni/Al_2_O_3_, which is in agreement with the formation of larger particles that could be easily reduced, as observed by H_2_-TPR. The average particle sizes of Ni for the used catalysts were in the range of 8–12 nm, which can be attributed to some sintering.

The EDS spectroscopy exhibited a uniform distribution of metal after doping with Ce, indicating an improvement in the distribution of metal in the samples. The EDS atomic composition showed an increase in the Ce and Ni contents compared with the XPS results. It was observed that the STEM images (Figure 8) of the distribution of Ni corresponded mostly to Ni^2+^ species and not to reduced species (lower peak intensity), as observed in Figure 5B and listed in Appendix A. At the surface, a higher Ce/Al ratio had a greater influence on activity, as shown in Appendix A. The best results were obtained in HTC 4, with the lowest Ni/Al ratio, for all tests of activity.

## 4. Conclusions

In this study, four different hydrotalcite materials derived via a co-precipitation technique with different Mg, Al, Ce, and Ni contents were investigated during the Sabatier process in the dark and under visible and UV LEDs, improving the performance of the monometallic Ni/Al_2_O_3_ in terms of the hydrogenation of CO_2_. The partial replacement of Al by Ce on the structure enhanced the activity at lower temperatures (<50 K) because the oxygen vacancies provided by CeO_2_ improved the adsorption of CO_2_ and the catalytic activity. The mixed oxide with a Ce/Al molar atomic ratio of 0.22 had improved photocatalytic activity at lower temperatures, achieving 50% CO_2_ conversion, 46% CH_4_ yield, and almost 99.5% CH_4_ selectivity under visible light at 473 K. When we compared monometallic Ni contents of 13% and 25% on γ-Al_2_O_3_, we observed that an increase in the metal content improved the thermal and dark activities in all temperature ranges, as expected. Furthermore, the thermal activity began at higher temperatures compared with the HTCs. Regarding the reduction temperature profiles of the catalysts, higher ranges of the reduction temperature for the spinel phases of the hydrotalcites exhibited better activity than the free NiO species or those weakly interacting with γ-Al_2_O_3_ at lower temperature ranges. The photocatalytic comparison between UV and visible light started with significant activity at low temperatures (473 and 523 K, respectively), with better results for visible light with the same light irradiance. In terms of viable application, these catalysts are low-cost alternatives and can be considered promising materials for the future photo-methanation of CO_2_ at lower temperatures.

## Figures and Tables

**Figure 1 materials-16-05907-f001:**
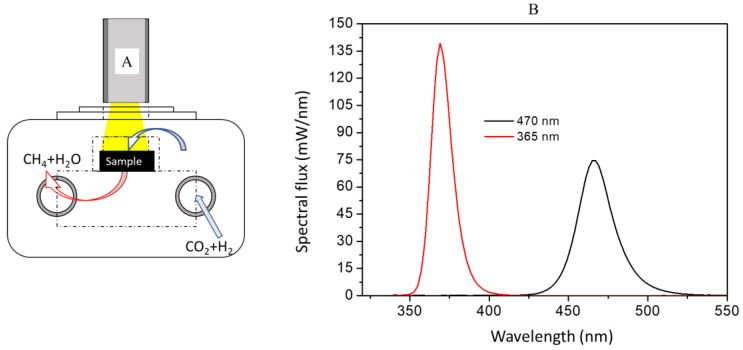
Scheme of the photoreactor (**A**). Spectral radiant flux of the LED lights (**B**).

**Figure 2 materials-16-05907-f002:**
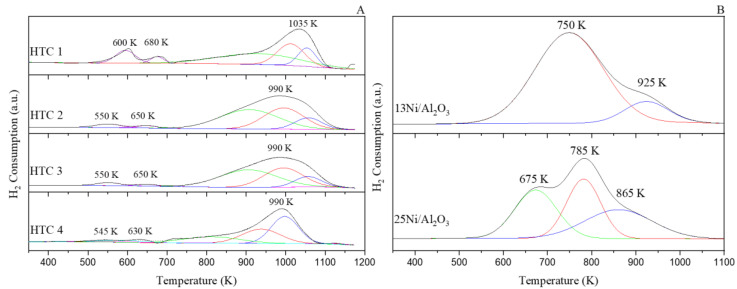
H_2_-TPR profiles of the calcined catalysts of (**A**) calcined HTCs and (**B**) Ni/Al_2_O_3_ catalysts.

**Figure 3 materials-16-05907-f003:**
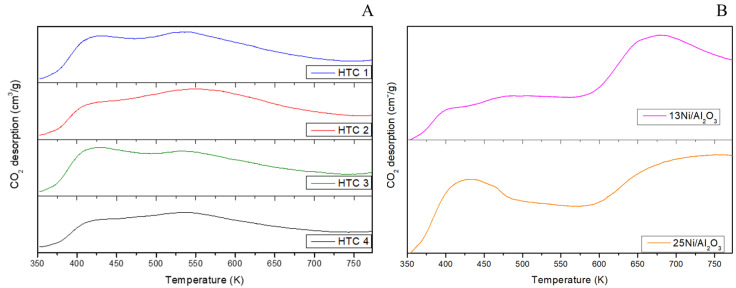
CO_2_-TPD profiles of (**A**) calcined HTCs and (**B**) Ni/Al_2_O_3_ catalysts.

**Figure 4 materials-16-05907-f004:**
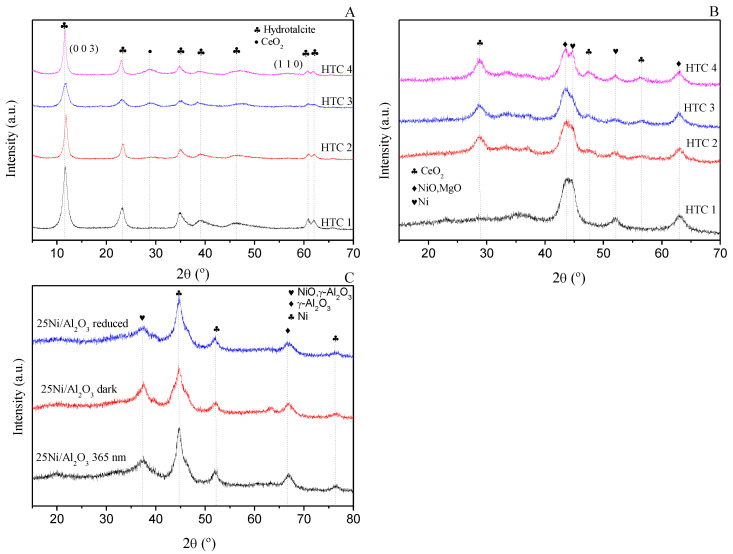
XRD profiles of hydrotalcites: (**A**) dehydrated; (**B**) reduced. (**C**) Comparison of the crystallinity of 25Ni/Al_2_O_3_.

**Figure 5 materials-16-05907-f005:**
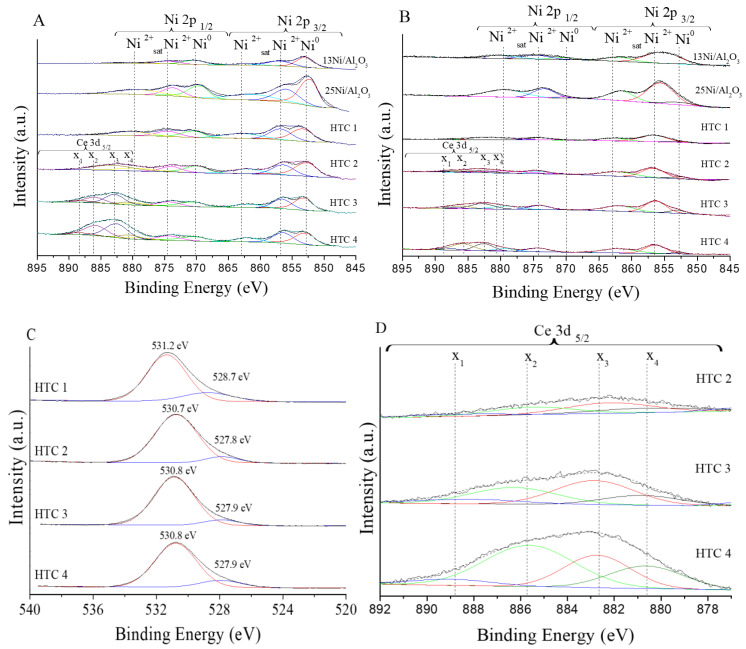
Comparison of the XPS spectra: (**A**) reduced Ni 2p; (**B**) used Ni 2p; (**C**) reduced O 1s; and (**D**) reduced Ce 3d. Different colors represent the deconvoluted peaks of each element.

**Figure 6 materials-16-05907-f006:**
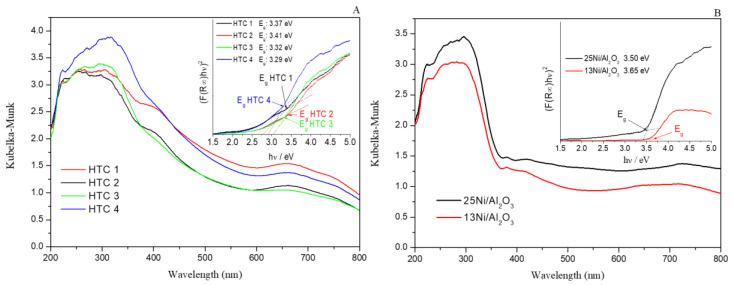
Diffuse reflectance UV–vis spectra of (**A**) hydrotalcites and (**B**) monometallic catalysts.

**Figure 7 materials-16-05907-f007:**
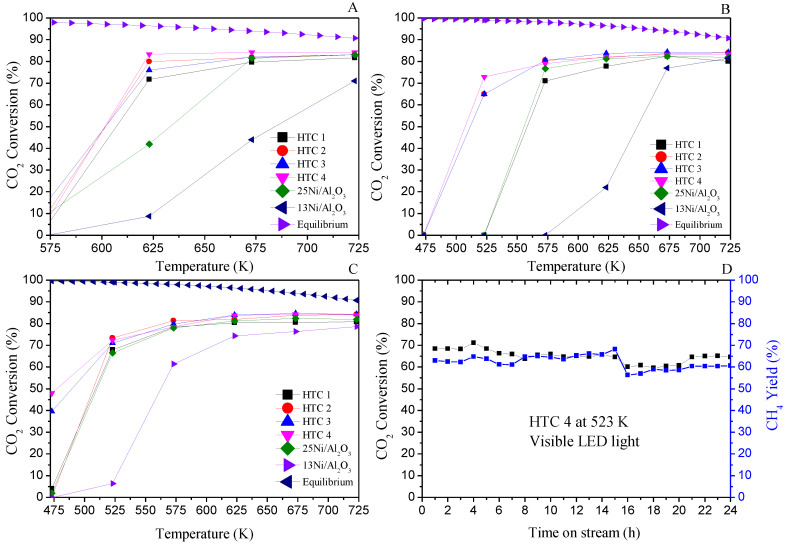
Comparison of activity enhanced under different conditions: temperature in the dark (**A**), UV light (**B**), and visible light (**C**). Stability of HTC 4 under visible light for 24 h at 523 K (**D**).

**Figure 8 materials-16-05907-f008:**
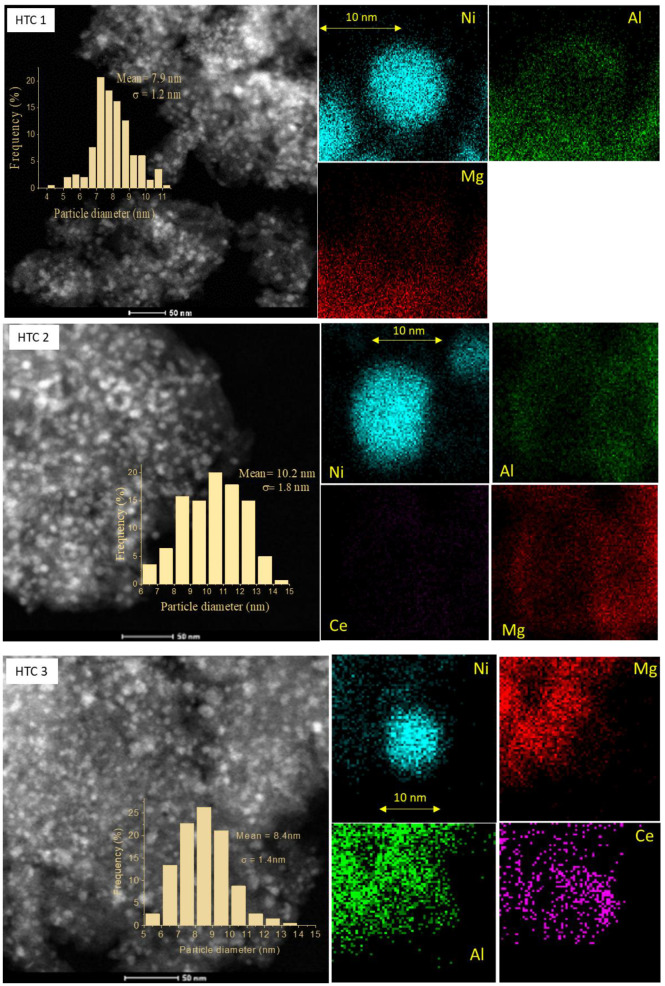
Scanning transmission electron microscopy (STEM) images of the used catalysts and the particle size frequency and energy dispersive X-ray spectroscopy (EDS) for the analysis of the surface dispersion of metal.

**Table 1 materials-16-05907-t001:** Summary of various catalysts and photocatalysts for CO_2_ methanation.

Sample	Metal Precursors	CO_2_ Conversion (%)	CH_4_ Selectivity (%)	T (K)	GHSV (h^−1^)	Stoichiometry	Flow (mL·min^−1^)	Refs.
Hydrotalcite	Al, La, Mg, Ni	46.5; 75	98; 99	523; 573	12	H_2_/CO_2_/Ar = 12/3/5	100	[13,14]
Hydrotalcite	Ni, Mg, Al	42–92	85; ≃100	523; 623	2400; 40–60	H_2_/CO_2_ = 4/1	40; 200–300	[2]
Hydrotalcite	Fe, Mn, Ni, Al	85–94	>99	523–573	200.000	H_2_/CO_2_/Ar = 4/1/5	150 NL h^−1^ g_cat_^−1^	[15]
Ni/CeO_2_ Ni/Al_2_O_3_ Ni/TiO_2_	Ni, Ce, Al	≃90 ≃57 ≃30	≃100 ≃97 ≃99	623	10	H_2_/CO_2_ = 4/1	100	[16]
Ni/Al_2_O_3_ Ni–La/Al_2_O_3_ Ni–Ce/Al_2_O_3_	Ni, Ca, La, Mg, Ce, Ba	86 99 95	≃99	673	38.3	H_2_/CO_2_ = 4/1	280	[17]
Rh/TiO_2_	Rh, Ti	5–16	>98	623	N/A	H_2_/CO_2_/Ar = 6.1/1.6/2.4	250	[18] ^a,b,c^
Ru/STO	Ru, Ti	89.5% in 1 h	12.6 mmol/(g·h)	423	44.6	H_2_/CO_2_ = 4/1	80	[19] ^d^
Au/TiO_2_ K–Au/CeO_2_ Au/CeO_2_	Au, Ti, Ce	20–50 5–50 5–38	N/A	573–773	3000–6000	H_2_/CO_2_ = 4/1	N/A	[20] ^f^
Ni/SiO_2_·Al_2_O_3_	Ni, Si, Al	94.9 ^e^	97.2	<423	N/A	H_2_/CO_2_/N_2_ = 7/1.5/1.5	N/A	[21]

^a^ LED light of 2.8 W cm^−2^; ^b^ no incident light; ^c^ LED light of 1 W cm^−2^; ^d^ 1080 W/m^2^; ^e^ solar simulator coupled with an AM1.5 filter; ^f^ 500 W medium-pressure mercury vapor lamp. N/A: Not available data.

**Table 2 materials-16-05907-t002:** The nominal loading was added to synthesize HTCs.

	Metal Content (wt.%)			
	Ni	Ce	Mg	Al
HTC 1	27	-	50	23
HTC 2	26	4	48	21
HTC 3	25	11	46	18
HTC 4	23	18	43	16

**Table 3 materials-16-05907-t003:** Summary of the metal contents and the textural properties.

	Composition (wt.%)						
	Ce	Ni	Mg	Al	S_BET_ (m^2^/g) ^a^	V_tot_ (cm^3^/g) ^b^	Dr (nm) ^c^
HTC 1		27.2	50.0	22.6	177.0	0.50	5.5
HTC 2	4.1	25.8	49.2	20.9	181.7	0.43	5.6
HTC 3	11.1	24.4	46.0	18.5	164.6	0.27	6.5
HTC 4	17.8	22.7	43.9	15.6	154.2	0.37	6.2
13Ni/Al_2_O_3_		12.6			178.3	0.33	6.6
25Ni/Al_2_O_3_		26.5			191.8	0.39	6.1

^a^ Surface area based on the BET equation. ^b^ Pore volume of BJH desorption. ^c^ Average pore diameter of BJH desorption.

**Table 4 materials-16-05907-t004:** Basic sites for the calcined catalysts.

Catalyst	Total Basicity (mmol/g)	Contribution of Weak Sites	Contribution of Moderate Sites	Contribution of Strong Sites
HTC 1	0.35	28.2%	47.2%	24.6%
HTC 2	0.35	23.2%	49.3%	27.5%
HTC 3	0.31	30.8%	44.0%	25.2%
HTC 4	0.35	21.9%	49.5%	28.6%
13Ni/Al_2_O_3_	0.17	13.2%	30.8%	56.0%
25Ni/Al_2_O_3_	0.34	15.2%	35.0%	49.8%

**Table 5 materials-16-05907-t005:** Crystal size of the non-calcined HTCs.

	Reflection (003)	Reflection (110)	Cell Parameter “c”
HTC 1 NC	12 nm	15 nm	2.34 nm
HTC 2 NC	8 nm	16 nm	2.34 nm
HTC 3 NC	8 nm	13 nm	2.34 nm
HTC 4 NC	13 nm	16 nm	2.36 nm

**Table 6 materials-16-05907-t006:** Results of H_2_ chemisorption.

Sample	Metal Dispersion (%)	Metallic Surface Area (m^2^/g Sample)	Active Particle Diameter (nm)
25Ni/Al_2_O_3_	2.02	3.36	50.1
13Ni/Al_2_O_3_	1.19	1.03	84.7
HTC 1	1.4	2.46	70.7
HTC 2	2.28	4.13	44.4
HTC 3	1.6	2.60	63.2
HTC 4	1.2	1.84	83.3

**Table 7 materials-16-05907-t007:** Metal/Al atomic ratios were obtained from the XPS results of the reduced and calcined catalysts.

	Reduced Catalyst XPS		Calcined Catalyst ICP	
	Ce/Al	Ni/Al	Ce/Al	Ni/Al
25Ni/Al_2_O_3_		0.163		0.307
13Ni/Al_2_O_3_		0.067		0.125
HTC 1		0.190		0.554
HTC 2	0.024	0.173	0.038	0.566
HTC 3	0.073	0.157	0.116	0.608
HTC 4	0.118	0.168	0.220	0.668

**Table 8 materials-16-05907-t008:** (**A**) Difference in CO_2_ conversion between dark and UV-irradiated conditions. (**B**) Comparison of activity in the dark vs. under visible light.

(A)	573 K	623 K	(B)	473 K	523 K
HTC 1	70	12.6	HTC 1	0	6
HTC 2	72.7	5.4	HTC 2	0	5.4
HTC 3	64.2	5	HTC 3	39.6	6
HTC 4	74	74	HTC 4	48	0
25Ni/Al_2_O_3_	66.3	45.8	25Ni/Al_2_O_3_	0	66.7
13Ni/Al_2_O_3_	0	13.3	13Ni/Al_2_O_3_	0	6.4

## Data Availability

The data presented in this study are available on request from the corresponding author.

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
