# Peer review of "Photo- and Thermocatalytic CO2 Methanation: A Comparison of Ni/Al2O3 and Ni–Ce Hydrotalcite-Derived Materials under UV and Visible Light"

_materials, 2023, doi:10.3390/ma16175907_

Round 1

Reviewer 1 Report

As a referee, I have the following concerns that must be addressed by the authors during a potential revision stage:

1- I preferred to easily understand the research gap and your study novelty by only reading the abstract.

2- All abbreviations must be defined at their first mention in the text.

3- all first three cited references are about CO2 methanation, not CO2 capture. It would be best if you highlight the advantages/disadvantageous of separation-based and conversion-based processes for CO2 removal.  

4- Avoid bulk citations and it is better to break them 

5- The current version of the manuscript reviews no literature. Hence, it is not possible to verify its novelty. You should completely review the literature covering the topic of your study. 

6- The difference between your study with respect to the following articles needs to highlight:

Kim, M.J., Youn, J.R., Kim, H.J., Seo, M.W., Lee, D., Go, K.S., Lee, K.B. and Jeon, S.G., 2020. Effect of surface properties controlled by Ce addition on CO2 methanation over Ni/Ce/Al2O3 catalyst. International Journal of Hydrogen Energy, 45(46), pp.24595-24603.

Feng, Y., Yang, W., Chen, S. and Chu, W., 2014. Cerium promoted nano nickel catalysts Ni-Ce/CNTs and Ni-Ce/Al2O3 for CO2 methanation. Integrated Ferroelectrics, 151(1), pp.116-125.

7- The combination of reaction and separation processes is also needed to mention as a viable technology for harmful gas removal using this article (Comparison of two different flow types on CO removal along a two-stage hydrogen permselective membrane reactor for methanol synthesis).

8- It is better to move Figures 1 and 2 to a more suitable location. They are not related to the introduction section. 

9- Why some cells in Table 1 are empty?

10- Provide valid reference(s) for all equations.

11- Do not use the dot symbol instead of the multiplication sign in equations.

12- I have no sense of Eqs. (5-7). Although all of them are dimensionless, the numerator and denominator are different!!!

13- Use the same font throughout the manuscript (i.e., text, figure, and table).

14- Some figures do not have enough quality.

15- Remove the gray background of Figure 9.

Reviewer 2 Report

The manuscript titled “Photo- and Thermocatalytic CO2Methanation: A Comparison of Ni/Al2O3and Ni-Ce Hydrotalcite-Derived Materials under UV and Visible Light” presented by Rafael Canales and V. Laura Barrio . However, there are major issues that could be clarified:

  • Abstract should not content abbreviations. In abstract authors state that the catalysts with Ce/Al=0.22 demonstrates highest activity, but what is the ratio Ce/Al they discuss (molar, weight, surface atomic)?
  • Introduction should be re-written, now it content much unnecessary information (about the reaction, hydrotalcite, etc).
  • The ICP-OES and BET data should be compacted and presented in the merged table, only Sbet(m2/g) is interested. The other low informative data should be moved to SI.
  • Could the authors explain H2-TPR study at temperatures higher then 723K (calcination temperature of catalyst)?
  • The authors used different terms to describe the catalysts under study (reduced and calcined, for example, Table 8). I should say that it is not easy to catch the author’s idea during the reading of manuscript. Please, re-write the manuscript omitting non-sensitive information, compacting the tables for more clear understanding.
  • At XPS part: the information about the scale (BE) correction is absent. Please, provide Al2p spectra in the SI.
  • The band gap for composite catalysts calculation is incorrect. See, doi/10.1021/acs.jpclett.8b02892. Please, provide for reference UV-vis spectra of HTC and Al2O3 w/o nickel.
  • The main idea of study was to demonstrate the influence of UV-vis irradiation. The section 3.2 does not allow authors to argue this idea. It necessary (at least) to demonstrate the measurement of rate of methane yield during the time at the given temperature.
  • The scheme at Fig. 10 is badly presented. Please, specify the band gap of composite catalysts and heterostructure.
  • The author state that “this is the first research that has evaluated the individualized impact of each metal in order to assess the behavior of each metal under different reaction conditions”. In fac, this is not correct statement, there is a lack of information about the structure of composite catalysts, the influence of oxides (NiO, NiAlOx, MgO, etc).

The manuscript demands some additional work. English should be revised.

English should be revised.

Round 2

Reviewer 1 Report

It seems the authors have forgotten to address my previous concerns and provide a relevant response to the comments.

Hence, I give them the last chance to thoroughly and appropriately address ALL my previous comments.

The English language is at an acceptable level. Minor editing of English language required.

Author Response

Please see the attachment file: "answer-reviewer MDPI Reviewer 1.docx"

Reviewer 2 Report

The authors revised and improved the manuscript. Now it could be accepted for publication. 

It is ok.

Round 3

Reviewer 1 Report

After two rounds of revision, it seems the authors do not want to appropriately modify the manuscript by completely addressing major parts of my concerns. Since I found their responses to the 1st, 3rd, 6th, 10th, 11th, 13th, and 14th comments incomplete, I have to suggest a major revision once again.
